# Machine Learning Methods for Automatic Segmentation of Images of Field- and Glasshouse-Based Plants for High-Throughput Phenotyping

**DOI:** 10.3390/plants12102035

**Published:** 2023-05-19

**Authors:** Frank Gyan Okyere, Daniel Cudjoe, Pouria Sadeghi-Tehran, Nicolas Virlet, Andrew B. Riche, March Castle, Latifa Greche, Fady Mohareb, Daniel Simms, Manal Mhada, Malcolm John Hawkesford

**Affiliations:** 1Sustainable Soils and Crops, Rothamsted Research, Harpenden AL5 2JQ, UK; 2School of Water, Energy and Environment, Soil, Agrifood and Biosciences, Cranfield University, Bedford MK43 0AL, UK; 3African Integrated Plant and Soil Science, Agro-Biosciences, University of Mohammed VI Polytechnic, Lot 660, Ben Guerir 43150, Morocco

**Keywords:** feature extraction, imaging, machine learning, phenotyping, segmentation

## Abstract

Image segmentation is a fundamental but critical step for achieving automated high- throughput phenotyping. While conventional segmentation methods perform well in homogenous environments, the performance decreases when used in more complex environments. This study aimed to develop a fast and robust neural-network-based segmentation tool to phenotype plants in both field and glasshouse environments in a high-throughput manner. Digital images of cowpea (from glasshouse) and wheat (from field) with different nutrient supplies across their full growth cycle were acquired. Image patches from 20 randomly selected images from the acquired dataset were transformed from their original RGB format to multiple color spaces. The pixels in the patches were annotated as foreground and background with a pixel having a feature vector of 24 color properties. A feature selection technique was applied to choose the sensitive features, which were used to train a multilayer perceptron network (MLP) and two other traditional machine learning models: support vector machines (SVMs) and random forest (RF). The performance of these models, together with two standard color-index segmentation techniques (excess green (ExG) and excess green–red (ExGR)), was compared. The proposed method outperformed the other methods in producing quality segmented images with over 98%-pixel classification accuracy. Regression models developed from the different segmentation methods to predict Soil Plant Analysis Development (SPAD) values of cowpea and wheat showed that images from the proposed MLP method produced models with high predictive power and accuracy comparably. This method will be an essential tool for the development of a data analysis pipeline for high-throughput plant phenotyping. The proposed technique is capable of learning from different environmental conditions, with a high level of robustness.

## 1. Introduction

There is an increasing need to improve agriculture to meet the 2050 global food demand agenda. One aspect of this improvement is the implementation of high-throughput plant phenotyping (HTPP) to enable the large-scale evaluation of plant performances in breeding programs. HTPP involves the automatic sensing of plant physical features to understand the dynamic relationship between plants and their environment [1]. This method could involve the use of non-invasive methods to collect plant data and aims to analyze plant physiology, phenology, and morphology by using sequentially acquired images from multiple sensors, including visible RGB, thermal, hyperspectral cameras, 3D laser scanners, and fluorescence imagers, taken from multiple view angles [2]. Digital RGB cameras are the most common and widely used image-sensing tools due to their relatively low cost, high resolution, and portability [3]. RGB cameras measure the light reflected from objects by using three broad bands centered at 650 nm (red), 550 nm (green), and 450 nm (blue) [4]. In processing digital images, segmentation is a critical step that separates and identifies regions of interest in images.

At the basic level, segmentation separates image pixels into two classes: foreground (i.e., plant vegetation) and background (i.e., non-plant objects). Over the years, several segmentation techniques have been developed, which can broadly be categorized as threshold, color-index, and learning-based methods [5]. Threshold-based methods separate image pixels into two groups based on their intensity values above or below a given threshold. This technique transforms an image from RGB to grayscale and binarizes the gray level based on the threshold value. The threshold is either global (using a single threshold value) or variable (threshold value varies). Binary, adaptive, and Otsu thresholding are examples of commonly used threshold-based segmentation techniques. Color-index-based techniques use combinations of color indices to differentiate plant vegetation from other non-plant objects. This method works on the assumption that the histogram distribution of an input image is bimodal and can be distinctively separated by a color threshold index. The color index of vegetation extraction (CIVE), excess green–red index (ExGR), and normalized difference index (NDI) are examples of commonly used color-based segmentation techniques [6]. Meyer and Camargo [7] developed an improved color index method (ExG-ExR) to segment plants (from the glasshouse) with field illumination conditions from their background. Kirk et al. [8] utilized a combination of color and threshold methods in estimating the leaf area index of cereal crops. Their method used a combination of greenness and intensity-derived indices from the red and green spectrum. Learning-based methods utilize supervised and unsupervised machine learning approaches to perform semantic and instance segmentation of plant images for disease detection [9], plant growth analysis [1], leaf counting [10], wheat spike counting [11], etc. Examples include k-means clustering, supervised mean-shift algorithm, fuzzy clustering, etc. Studies have shown that machine learning models, when trained with carefully selected features, outperform the segmentation methods described above. Adams et al. [12] utilized a supervised learning method to segment plants by classifying pixels-based datasets whose labels were produced using a k-means clustering algorithm. Their results showed the better performance of their approach as compared to traditional thresholding methods. In recent times, deep learning, a branch of the learning-based method, has been extensively applied to segment plants with different backgrounds. A commonly used deep learning architecture called the convolution neural network has gained much attention in computer-vision-based plant phenotyping. Deep learning approaches for the semantic segmentation of plants, including R-CNN, pyramid networks, and encoder–decoder-based models, rely on the concept of CNN modelling [13]. These methods have been extensively used in plant organ segmentation for root and shoot phenotyping. These include the works of Aich and Stavness [10], who used SegNet architecture, a deconvolutional network without a fully connected layer to segment plant leaves from its background. Moris et al. [14] also performed instance segmentation to detect and discriminate leaf boundaries, using a pyramid convolution neural network. Recently, Narisetti et al. [15] developed a fully automated segmentation pipeline based on pretrained U-net deep learning models to phenotype shoots of different crops. Although deep learning methods have been extensively used in plant segmentation and phenotyping, they require a large-volume labelled dataset, which is not readily available [16]. Furthermore, deep learning models have a high number of trainable parameters (e.g., Resnet18 has around 11 million, while denseNet has around 8 million trainable parameters), resulting in the slow processing of data, particularly when dealing with high-throughput data. This forms a bottleneck around plant phenotyping, especially when dealing with a small number of plants datasets. Due to these limitations, deep learning approaches were not considered in this study.

Many segmentation methods rely on color information [17], which is a strong descriptor used in object detection and recognition. However, color-based segmentation may not perform well in certain scenarios, such as cluttered backgrounds, variable illumination conditions, etc.). Plant color can vary depending on the position of the sun during image acquisition [18], which can lead to poor segmentation results due to the change in color caused by different illumination conditions. Using different color spaces can minimize the effects of illumination and color variability on segmentation. Color spaces provide additional information about pixel intensity and color distribution and can be used individually or in combination to segment plant images [19]. Prasetyo et al. [20] combined different color spaces to segment and detect mango trees. They transformed the RGB color space to HSV (Hue Saturation Value) and YCbCr (luminance chrominance), where individual color channels were combined with Otsu thresholding for segmentation. Their results showed Cr from YCbCr as effective at separating mango trees from the background. Similarly, Sadeghi-Tehran et al. [3] used multiple color features obtained from different color spaces to train a supervised machine learning classifier for wheat segmentation in field images. They found that using multiple color spaces improved segmentation accuracy and was more robust to background noise and illumination changes compared to using a single-color space.

Although the use of different color spaces for segmentation can be effective, it also can be computationally expensive, particularly when computing millions of pixels, and requires high computer speed and large memory capacity for processing. In addition, some of the individual color channels may have redundant effects on the segmentation process, while others may have little or no effect. Furthermore, this method may fail to properly segment plants with different shades of color and texture across different parts of the plant image if the appropriate color features are not utilized. This study sought to (1) develop an automated phenotyping pipeline based on the neural network to segment field- and glasshouse-based plant images for high-throughput plant phenotyping and (2) predict plant SPAD values based on the proposed and other existing segmentation methods.

## 2. Results

For pixel-wise plant segmentation, plant color features were extracted and used as inputs for training a multilayer perceptron neural network. The features belonged either to the AF (all features) dataset which contains all the transformed color features; or the SF (selected features) dataset which contains only the color features after feature selection.

### 2.1. Performance of the ML Models Based on Different Number of Training Features

Figure 1 depicts a whisker plot summarizing the accuracy scores of the models. Table 1 also summarizes the classification scores of all three models. With the MLP model, the SF datasets outperformed the AF dataset (99.321% vs. 98.464% accuracy). A similar trend was observed in the RF model, with a 0.343% difference between the SF and AF datasets accuracy. However, in the SVM model, the mean accuracies of the SF and AF datasets were statistically non-significant (*p* > 0.05). As expected, all the models trained with the SF dataset had shorter processing times than those trained with the AF dataset. For instance, during training, the processing speed for the SVM model significantly decreased from 560.560 s in the AF dataset to 321.502 s in the SF dataset. Similarly, during testing (Appendix A), the SF models had shorter processing times (2.450 s, 44.81 s, and 86.163 s for the MLP, RF, and SVM models, respectively) than the AF-dataset-trained models (5.150 s, 81.225 s, and 132.158 s for the MLP, RF, and SVM models, respectively) when tested with 10 image datasets. This shows that the selected features did not only improve the performance of the classification models but also reduced the processing time by using important and informative features for training the models. For this reason, all the three models were trained with the SF datasets for segmentation in the subsequent analysis.

### 2.2. Segmentation Accuracy Analysis

Segmentation accuracy was evaluated using three quality factors, namely Qseg, Sr, and an error factor (Es), as explained in Section 4.5. Figure 2 shows examples of segmented wheat and cowpea images, using the MLP and ExG methods. By visual inspection, the proposed method produced high-quality images as compared to the traditional ExG segmentation method.

Moreover, as shown in Figure 3, the ML-based methods (MLP, SVM, and RF) outperformed the color-index methods (ExG and ExGR) for glasshouse-based cowpea plants. The MLP had the highest average Qseg (0.916) and Sr (0.956), with the lowest standard deviations (0.012 and 0.040, respectively), indicating less variability in the Qseg and Sr scores. The RF model followed closely, achieving 0.874 (Qseg) and 0.932 (Sr), with standard deviations of 0.046 and 0.044, respectively. ExG had the lowest Qseg and Sr values (0.785 and 0.816), with the highest Es (0.220) indicating more misclassification of pixels. Again, MLP had the lowest classification error (0.058), followed by RF and SVM (0.074 and 0.170, respectively).

The results presented in Figure 4 shows that the color-index-based methods (ExG and ExGR) had the lowest Qseg values (0.725 and 0.693, respectively), with high standard deviations (0.082 and 0.112, respectively) for field-based wheat images. On the other hand, the MLP method had the best performance, earning the highest Qseg rating (0.880), with a low standard deviation (0.055). This indicates that the MLP model is the best model, producing better pixel classification with an error rate of 0.128 as compared to the other techniques. In both environments, i.e., the glasshouse and the field, the ML-based methods outperformed the color-based methods, and this is consistent to the finding of Adams et al. [12], who suggested that ML-based techniques are an improvement over traditional segmentation methods.

### 2.3. Performance of SPAD Prediction Regression Models

The root mean square error (RMSE), the adjusted determination coefficient (R^2^), and the mean absolute error (MAE) were the metrics used to comparatively assess the performance of the SPAD regression models.

#### 2.3.1. Cowpea SPAD Values Prediction

Table 2 shows the list of extracted CVIs with their corresponding correlation coefficients (r) for cowpea. The correlations of the measured SPAD values and the extracted CVIs were influenced by the quality of the segmented images; for example, poorly segmented images with high pixel misclassification and noise led to lower correlations with the measured SPAD values and vice versa. The CVIs extracted using the MLP segmentation method had high correlation coefficients, with DGCI having the highest (0.948) and EGI having the lowest (0.464) coefficient. This was followed by the RF segmentation method, with the highest correlation (0.850) recorded with the DGCI index and lowest (0.295) recorded with the EGI color vegetation index. The ExG segmentation method had the lowest correlation coefficient (0.228) recorded for the blue CVI whiles its highest correlation coefficient (0.742) was recorded with ExG CVI.

Table 3 shows the results of the regression models for predicting SPAD values for the glasshouse-based images. The MLP-GR model had the highest R^2^ value in both the training and testing phases, while the ExG-GR model had the lowest. The R^2^ values for both training and testing datasets were close, indicating that the models did not underfit nor overfit the datasets. The RMSE has an inverse trend with the R^2^ metric (the lower the RMSE, the better). Figure 5 is a scatterplot of observed and predicted SPAD values from test data which shows that the MLP-GR model with a high R^2^ (0.904) had the best prediction performance, followed by the RF-GR model (R^2^ = 0.837), while the ExG-GR model had the lowest prediction power (R^2^ = 0.701). Overall, all the models performed well, with high prediction power (R^2^ > 0.7) in both the training and testing phases.

#### 2.3.2. Wheat SPAD Values Prediction

Table 4 shows the Pearson correlations coefficient between the selected CVIs and SPAD readings. The MLP segmented method shows a high correlation with the SPAD values for all the extracted CVIs, with the highest observed for ExG CVI (0.927) and the lowest for EGI CVI (0.412). The ExG segmentation method yielded relatively low correlations, with the highest correlation for ExG CVI (0.704) and lowest for EGI CVI (0.111), indicating poorly segmented images. Among the extracted CVIs, ExR, CIVE and DGCI demonstrated high correlations with all the segmentation methods, which is consistent with the finding of Hassanijalilian et al. [25]. Table 5 shows the performance of the regression models for the training and testing datasets. The MLP-FR model exhibited the best performance, achieving the highest R^2^ score for both training (0.893) and testing (0.821) datasets, with the lowest MAE scores (2.520 and 3.233, respectively). This indicates that the MLP method segmented the images well, with minimal errors. RF-FR and SVM-FR followed with R^2^ = 0.815 and 0.791, respectively, for training, and 0.775 and 0.747, respectively, for testing. The ExG-FR model had relatively the lowest performance, with R^2^ = 0.642 and 0.521 for training and testing, respectively. This could be attributed to the poorly segmented images obtained using the ExG method, resulting in many misclassified pixel features for training the regression model. Figure 6 shows the linear regression between the predicted and observed SPAD values for the different segmentation methods using the wheat testing dataset. The MLP-FR model demonstrated better prediction accuracy (R^2^ = 0.821), outperforming all the other methods, while the ExG-FR model had the lowest prediction accuracy (R^2^ = 0.546).

## 3. Discussion

Images of plants acquired from environments with varying conditions present daunting tasks for accurate segmentation of the foreground and background. This is due to conditions such as illumination variability and different coloration of plant leaves resulting from factors such as nutrient deficiencies. These challenges are particularly predominant in field-based images, where there exist other complex situations, such as cluttered backgrounds and leaf occlusion, resulting in the underestimation of vegetation pixels within the images. While color-index-based segmentation is a widely used low-level pixel feature for images with uniform background and contrast, it is not effective for images with complex backgrounds or canopies with non-uniform color distribution. In this study, the color-index methods failed to adequately segment vegetation, resulting in multiple fragments of the plants due to color disparities. However, ExGR demonstrated good segmentation quality for images taken from the glasshouse and field compared to ExG. This is in agreement with Guijarro et al. [26] and Meyer and Neto [7], who found that ExGR and combinations of other color indices produce improved segmentation quality compared to the ExG method.

A single-layer MLP network was proposed that takes in annotated and color-transformed pixels and outputs binary classified pixels. The annotation covered all plant growth stages to ensure that the model effectively learns different scenarios of plant color characteristics, such as yellow-to-brown color due to nitrogen deficiency and deep green color from healthy plant leaves. The MLP model with a high training and testing accuracy produced quality segmented images with reduced noise levels. In a similar study, Bakhshipour et al. [27] used an artificial neural network (ANN) trained with wavelet texture features to achieve a 96% accuracy in discriminating weeds from sugar beets in the field. This supports the idea that neural networks such as MLP and ANN can successfully segment plant images when trained with the right phenotypic traits. The results show that the proposed MLP segmentation model is robust, achieving a high segmentation accuracy when used on plants from different environment scenarios and with complex leaf architecture and occlusion. Furthermore, the feature selection technique employed improved the robustness of the model by reducing the processing time and ensuring that the model was trained on only sensitive and non-redundant features. With a reduced number of training features, the MLP processing time was reduced by 62.8%, whilst the SVM and RF processing time was reduced by 49.1% and 57.4%, respectively. This is important in real-time processing, where high computational speed is required for the batch processing of images. Although the proposed method by Sadeghi-Tehran [3] successfully segmented plant vegetation from cluttered backgrounds, the large-features dataset used for model training presented a computational challenge. Unlike the proposed method, a high-powered computer memory is required to implement such segmentation techniques for the automated processing of images. Furthermore, although deep learning networks such as convolution neural networks (CNNs) have been successful in plant segmentation [23,24,25,26,27,28], they require large datasets to train and validate the model to ensure better performance. However, in this study, only a small dataset (less than 10% of the total dataset) was used to train the neural network for segmentation. Moreover, due to the fast nature and less training data requirement of the proposed method, it can easily be deployed on edge devices such as mobile phones, Raspberry Pi, and other devices as compared to CNN-based segmentation models.

Generally, ML-based methods have been successful in segmenting plant vegetation from complex environments under varying illumination conditions compared to color-index-based methods. The additional pixel information resulting from the color space transformations enhances the discriminatory power of the segmentation model. Supervised ML models demonstrated improved segmentation quality, which is consistent with the finding of McCauley et al. [29], who suggested that ML techniques that learn color features from multiple color orientations improve the robustness of the segmentation model. While the SVM model performed well, its high computational speed and memory requirements make it less feasible for high-throughput plant phenotyping.

After segmenting the images, machine learning regression models were developed based on the different segmentation methods to predict plant SPAD values. Generally, the method with good segmentation quality produced high prediction accuracies. In this case, the MLP segmentation method outperformed the others, achieving a high R^2^ (0.904, 0.821) and low RMSE (2.862, 3.680) for both greenhouse and field plants. This is in line with the Qui et al. (2021) study, which asserts that neural networks have high potential in estimating the nitrogen nutrition index (NNI) in rice compared to other statistical and machine learning algorithms. They developed six machine leaning models, namely SVM, ANN, RF, k-nearest neighbor (KNN), partial least square regression (PLSR), and adaptive boosting (AB), based on rice RGB images from an unmanned aerial vehicle (UAV). Their results show that ANN can estimate the NNI at the early jointing and filling stages of rice with R^2^ = 0.72 and 0.80, respectively. While Hassanijalilian et al. [25] achieved a similar performance in predicting soybean chlorophyll by using a SVM model (R^2^ = 0.86 and RMSE = 3.20), their method was based on a few randomly selected leaves which may not represent the actual scenario of chlorophyll dynamics in the field. Moreover, their object-recognition method (segmentation) is only feasible for small plant datasets.

Overall, the MLP method achieved the best performance in all the conditions tested in this study. It produced the highest segmentation quality with lower variations. The MLP method is versatile and can be used to segment not just green vegetation in the glasshouse but also field-based plant leaves with variable colors resulting from, for example, a nutrient deficiency or senescence.

## 4. Materials and Methods

Experiments were set up in two different environments: glasshouse (cowpea) and field (wheat). The segmentation pipeline, as shown in Figure 7, involves (1) image acquisition from two different environments, (2) image annotation and color transformation, (3) feature extraction and selection, and (4) modelling of the machine learning classifier.

### 4.1. Experimental Setup and Imaging

#### 4.1.1. Cowpea Grown in the Glasshouse

*Vigna unguiculata* (cowpea) plants were grown at the Plant Growth Facility (https://www.cranfield.ac.uk/facilities/plant-growth-facility, accessed on 10 March 2023) located at Cranfield University (Cranfield, UK). The plants were grown in pots filled with compost that was washed and depleted of macro- and micronutrients. The compost was then reconstituted with prepared nutrient solutions of defined composition. The compost-washing method followed the method by Masters-Clark et al. [30], with some modifications. This was achieved by flooding one-part compost to five-parts deionized water, mixing and breaking aggregates. The soluble solution was drained through a double-headed 0.5 µm sieve. The steps were repeated five times, and the depleted-nutrient compost was dried in an oven at 70 °C for 72 h. After drying, the compost was mixed with a nutrient solution prepared according to the treatment required. Four nutrient solutions were prepared based on Letcombe nutrient solution [30], with a high level of nitrogen and high level of phosphorus (HNHP), high level of nitrogen and low level of phosphorus (HNLP), low level of nitrogen and high level of phosphorus (LNHP), and low level of nitrogen and low level of phosphorus (LNLP). The concentrations were 49.1 mM and 14.6 mM N for HN and LN, and 13.4 mM and 3.3 mM P for HP and LP, respectively. Each treatment was replicated five times. The different treatments induced changes in leaves’ pigmentation, resulting in variations in the spectral characteristics (color) during plant development. During their development stages, plants with adequate nitrogen levels exhibited green leaves, whilst deficiency was characterized by chlorosis progressing from light green to yellow to brown, and phosphorus deficiency inhibited shoot growth and decolorized leaves from a blue-green color to pale green/yellow in severely affected regions [29]. Digital images were taken twice weekly, using a high-resolution (3296 × 2472 pixels) RGB camera (color 12-bit Prosilica GT3300) installed on the Lemnatec phenotyping platform in the glasshouse. The camera was set 3.5 m above the top plant canopy, covering an area of 0.48 m^2^ during imaging. Imaging commenced at the vegetative growth stage, 28 DAS (days after sowing), when the first trifoliate leaf had unfolded at node 3 and ended at the physiological maturity stage (90 DAS), when the seed pods were matured and ready for harvesting. The format of all the images used in this study was JPEG.

#### 4.1.2. Wheat Grown on the Field

Wheat (*Triticum aestivum), cv* Chilham, was cultivated in the field at Rothamsted Research, Harpenden, UK, from October 2020 to August 2021. Rows were planted 15 cm apart, with 350 seeds/m^2^ sowing density, on a 9 m × 1.8 m plot. The soil phosphorus (P) treatments were established by successive cropping and differential P application over several years at 12 Olsen phosphorus rates (approximately 3, 6, 9, 12, 15, 18, 21, 25, 30, 40, 50, and 60 ppm). The phosphorus deficient treatments induced color variations, especially on the leaves, veins, and stems, resulting in a purple color [31]. Digital images were taken weekly, starting from the jointing to booting stage, producing over 200 images. To capture these color variations, a digital camera -Sony Alpha a6300 (Sony Electronics, Sony UK, Weybridge, (Surrey), UK) was used. The camera was mounted on a 1.5 m monopod [32] and held perpendicular to the ground, approximately 2 m above the canopy. A cable release trigger (UR-232-Tasman) was attached to the monopod to control the camera trigger. The setup was centered in the middle of the plots. The camera was set to automatic exposure mode to compensate for the changes in natural light.

### 4.2. White Balance and Color Correction

Changes in light intensity and brightness throughout the day can impact the appearance of plant color in images, potentially leading to inaccurate interpretations [33]. To address this issue, we utilized color normalization/correction techniques. The goal of color correction is to transform image pixels, such that the pixel values of a reference (in this case, color checker tiles) match with the transformed images [34]. To accomplish this, we compared the pixel values of the captured images to the true colorimetric values of a color checker passport (X-Rite Pantone, Manchester, UK), which contains 24 squares of colors ranging from gray to brown [35]. The gray squares were used to establish the white balance, while the green, brown, and yellow squares were used to calibrate the corresponding hue values. The calibration process identified how much the hue value of the green, brown, and yellow squares on the color checker deviated from standard hue values, and the correction was then applied to the plant images.

### 4.3. Machine Learning Segmentation Process

#### 4.3.1. Image Annotation and Color Transformation

During the image-processing stage, illumination differences can create difficulties in achieving accurate segmentation. Various color spaces have been shown to possess unique characteristics that can minimize the impact of illumination on images and can resist noise variations during processing. Based on this concept, color features were extracted from different color channels to build a segmentation model. To accomplish this, image patches were created for plants at various growth stages and non-plant materials, which were manually annotated [36] as the foreground and background (Figure 8). The foreground consisted of 1400 patches (800 from field and 600 from glasshouse images), while the background had 1100 patches (600 from field images and 500 from glasshouse images). Each patch was then resized to a 20 × 20 pixels patch and transformed into eight different sets of color channels, including RGB, HSV, YCbCr, Lab, YUV (Y = luma or brightness, U = blue projection, and V = red projection), Luv (L = luminance, u = blue axis, and v = red axis), hls (h = hue, l = lightness, and s = saturation), and XYZ (chromatic coordinates). In this case, each pixel with eight transformed color channels gives a final feature vector of size 24. Each patch represents a *m x n* matrix, where *m* is the number of pixels, and *n* is the number of color channels. Hence, for each image patch, (20 × 20) × 24-pixel features were obtained. With 1400 foreground and 1100 background patches, a total of 24,000,000-pixel features (foreground = 13,440,000, and background = 10,560,000) were used to train a binary classifier for segmentation.

#### 4.3.2. Feature Extraction and Selection

Training a binary classifier with many pixel features can be challenging due to high computational memory and speed requirements. Additionally, some features were observed to be highly correlated (Figure 9a) and redundant, leading to similar effects during training. To address these issues, a feature-selection method called decision-tree-based recursive feature elimination (DT-RFE) was applied. DT-RFE is a wrapper-type feature selection method that uses a backward feature illumination technique to remove features in order of importance. It works on the principle of decreasing the size of a feature set by recursively selecting the most important ones. The method works by first training the decision tree model with a set of features. A significance mark of each feature is obtained, and weights are assigned to each feature. Then, a ranking criterion of all features is computed by using the sum of the weights of each feature in the model [37]. The features are then sorted according to the final score (weighted sum), and the smallest ranking features are removed. The process repeats itself until the desired number of highly sensitive features are achieved [38,39].

In this study, 12 features with an importance score above a 0.04 threshold were selected as input features for training the classifier (Figure 9b). This threshold was selected based on a trial experiment with different feature combinations. It was observed that a combination of the first 12 selected channels (with a high importance score) resulted in high accuracy of classification, whilst additional channels had no impact on overall accuracy. Moreover, when fewer than 12 channels were used, the classification accuracy dropped.

The important features selected for training the binary classifier are shown in Figure 9b. This includes the R and G (from RGB), H and S (from HSV), L and a (from Lab), v (from Luv), y (from ybr), u (form YUV), h, l, and s (from hls). These selected channels have properties such as high contrast, illumination invariant, and color differentiation that help in accurately separating plant pixels from non-plant pixels. For instance, the H and S channels separate the color information from intensity, reducing the effect of illumination and shadows on the robustness of the segmentation method. Additionally, the L and a components differentiate varying shades of color, with L indicating lightness, and a showing the red/green coordinate, which is dominant in the color makeup of plant pixels. Color transformation channels and their impact on image processing are highlighted in the work of Kumar and Miklavcic [40].

#### 4.3.3. Training and Testing the Model

An artificial neural network was utilized to perform the plant image segmentation, using a typical MLP architecture. MLP is a supervised neural network that maps a set of input features to a set of output in a feedforward propagation manner [41]. It is made of a set of processing elements (neurons) that forms different layers [42]. It has three main layers: input layer, one or more hidden layers, and output layer. The input layer takes in the values of the input data, while the output layer has one unit for each value of the MLP’s output. (It has a single unit for regression or binary classification and M units for M-class classification.) The hidden layers are fixed between these two layers and are fully connected, transmitting information to the output layer. The neurons in the different layers are interconnected to each other through weights whose values are updated after a learning process [43]. The weights define the importance of the interactions between the neurons in order to control the errors in the output layer [44]. Inputs from the last hidden layer converge to a single value in the output layer [45].

The MLP is trained with a dataset X number of inputs: X = (x_1_, x_2_, x_3_, x_4_…… x_n_) and gives y number of outputs Y = (y_1_, y_2_, y_3_…… y_n_), using a linear or nonlinear function f(.): R^x^ → R^y^ [46] to solve classification and regression problems. The success of an MLP network is dependent on parameters such as the activation function and loss error, which can be expressed mathematically as a linear combination, as shown in Equation (1) [47]:(1)$$P=\alpha_{0}[\sum_{j=1}^{n}\left({Wk}_{j}\alpha_{h}\sum_{j=1}^{m}\left(W_{ji}X_{i}+W_{jb}\right)\right)+{Wk}_{b}]$$
where P is the predicted output; n is the number of output neurons; Wk_j_ is the weight for the connecting neuron of hidden and output layers; α_h_ is the hidden neuron’s activation function; and m is the number of hidden neurons; W_ji_ is the weight for the connecting neuron of input and hidden layers; X_i_ is the input variable; W_jb_ is the bias for the hidden neuron; Wk_b_ is the bias for the output neuron [48]; and α_h_ and α_0_ are the activation functions for the hidden neuron and the output neuron, respectively. The activation function is a key part of a neural network. It is a mathematical expression that defines how the weighted sum is transformed from one layer to another. Basically, it tells whether a node should be activated or not. It is calculated from the product of the weight and input and adding a bias. Commonly used activation functions include linear, sigmoid, rectified linear unit (ReLU), etc. The activation function is usually linear for regression problems and sigmoid or rectified linear unit for classification problems [47]. The default activation function for the hidden layers is ReLU.

In this study, we employed a modified version of the MLP network used by Sari and Kefari [49]. This network was trained on the principle of back propagation algorithm [50]. This algorithm involves the forward propagation of data to calculate the loss between the target and predicted value, followed by the backward propagation through weight adjustment to reduce the loss [51]. The weights are updated after each iteration based on the expression in Equation (2):(2)$$W_{u}*=W_{o}-a\;(\frac{\partial Error}{\partial Wo})$$
where W_u_* is the updated weight, W_o_ is the old weight, a is learning rate, and ∂Error is the derivative of error with respect to the weight.

A preliminary study was conducted to select the number of hidden layers with the 12 input features per pixel. We tested 1 hidden layer (with 3 neurons), 2 hidden layers (3 and 10 neurons), and 4 hidden layers (3, 5, 10, and 20 neurons), and the preliminary results showed that there was no significant difference in the pixel classification when the number of hidden layers was increased. Hence, a grid search cross-validation (k = 10) was conducted to select the number of nodes in the 1 hidden layer. MLP was experimentally set to 12, 20, 35, 60, and 100 neurons. ReLU was used as the activation function for all processing nodes of the MLP model [52].

### 4.4. Regression for SPAD Values Prediction

SPAD readings are used to determine the greenness of plants, and this is highly correlated with their nitrogen content [53]. In this study, the plant chlorophyll index was measured using a SPAD meter (SPAD-502, Konica Minolta Sensing, Osaka, Japan) at different growth stages of wheat plants. For the field experiment, three readings were taken from five randomly selected wheat leaves within each experimental plot. SPAD readings were taken on the first, second, and third sectors of each fully expanded leaf. The final reading for each plot was the average of the fifteen measured SPAD values. In the glasshouse experiments, SPAD readings were taken concurrently with the image acquisition from cowpea leaves. Three readings were taken from fully expanded leaves located at the top, middle, and bottom part of each plant in each pot. The final plant SPAD values were calculated as the average of these readings.

The accurate segmentation of plant images is crucial for the precise and non-destructive measurement of plant traits. Poorly segmented images can result in higher levels of pixel misclassifications during segmentation, which may ultimately affect the analysis of plant traits. To evaluate the performance of the segmentation methods used in this study, we implemented a gradient-boosting decision trees (GBRTs) algorithm called eXtreme Gradient Boosting (XGBoost) as a regression model to predict SPAD readings of plants using the segmented image data. XGBoost is an ensemble machine learning algorithm for classification or regression predictive modeling problems [54]. It is computationally efficient and highly effective for structured datasets on classification and regression predictive modeling problems [54]. XGBoost builds regression trees one by one such that the subsequent tree is trained from the residuals of the previous tree. That is, the new model (tree) improves on its performance by correcting errors made by the previously trained model (tree) [55]. To build an XGBoost model, an initial tree (model) X_o_ defined to predict a target variable y is associated with a residual (y-X_o_). A new model, X_1_, is fit to the residuals from the previous model. In the next step, the previous model is combined with the new model X_1_ to produce a boosted version of X_o_. In this case, the mean squared error from X_1_ will be lower than that from X_o_. To improve the performance of X_1_, the model could be boosted by modeling after the residuals of X_1_ to create a new model X_2_. This process can be repeated until the residuals have been reduced as much as possible. In this work, the XGBoost implementation uses a histogram-based split finding method, which groups features into discrete bins to improve the training speed and reduce memory usage. The hyperparameters of the XGBoost, including n_estimators (the number of trees used in the model), subsample size used to train each tree, maximum depth (max-depth) of trees, and learning rate(α)- (indicating how fast the model learns), were fine-tuned to optimize the performance of the models.

The segmentation algorithms and extraction of plant traits were performed in the python 3.7 environment (python software foundation, http:www.python.org/, accessed on 1 August 2022) with scikit-learn, opencv2, and SciPy packages. The entire machine learning process was carried out on an intel core computer with a 2 GHz processor, 8.00 GB of RAM, and MS Windows 10 operating system.

### 4.5. Comparing Segmentation Performance

The performance of the MLP segmentation method was compared to the two ML-based segmentation methods and two well-known color-threshold methods: ExG (excess green) and ExGR (excess green–red). ExG is a simple color-index-based method that provides a clear distinction between soil and plant vegetation by increasing the contrast of the green part of the spectrum against the red and blue parts [56]. It operates on the bimodal distribution of pixels and is given by the expression ExG = 2 × G − R − B, where R, G, and B are the pixel values of the red, green, and blue color channels, respectively. ExGR was introduced by Meyer et al. [57], where two color indices, ExG and excess red (ExR), were combined to reduce the residual background pixels, achieving a high segmentation accuracy. ExGR is expressed as ExG − (1.4 × R − G), where R and G are the mean values of the red and green channels, respectively, and ExG is the excess green index. Both ExG and ExGR indices were combined with automatic Otsu thresholding to segment the images.

To evaluate the performance of the different methods, ten images each for glasshouse and field plants were randomly selected and segmented using all the five segmentation methods. The performance of all the methods was evaluated with reference images in which plant vegetations were manually segmented from the ten plants, using Adobe Photoshop Software (Adobe Inc., Maidenhead, UK). Figure 10 shows two examples of the test images and the manually segmented reference images for glasshouse and field conditions. Three quality factors, namely Qseg [7], Sr [58], and an error factor (Es) [59], expressed in Equations (3)–(5) were used to evaluate the quality of the segmentation methods.
(3)$$\mathrm{Qseg}=\frac{\sum_{x,y=0}^{x,y=h,w}\left(S\left(p\right)x,y\cap R\left(p\right)x,y\right)}{\sum_{x,y=0}^{x,y=h,w}\left(S(p)x,y\cup R(p)x,y\right)}$$
(4)$$\mathrm{Sr}=\frac{\sum_{x,y=0}^{x,y=h,w}\left(S\left(p\right)x,y\cap R\left(p\right)x,y\right)}{\sum_{x,y=0}^{x,y=h,w}\left(R(p)x,y\right)}$$
(5)$$\mathrm{Es}=\frac{\sum_{x,y=0}^{x,y=h,w}\left(S\left(p\right)x,y\cap R\left(!p\right)x,y\right)}{\sum_{x,y=0}^{x,y=h,w}\left(R(p)x,y\right)}$$
where S is the segmented plant, (S(p) = 255), or background pixels (S(p) = 0); R is the reference image; indices h and w are the height and width of the image, respectively; and x and y are the pixels’ coordinates. The segmentation accuracy is based on the logical operations logical or (∪), logical n (∩), and logical not (!). It is compared on a pixel-by-pixel basis of the segmented image (S) to the reference image (R).

Qseg measures the consistency on a pixel-by-pixel basis between the S and R images and ranges between 0 and 1, where 1 is perfect segmentation and vice versa. Similarly, Sr measures the consistency of plant pixels between image regions [59]. The Es measures the section of misclassified plant pixels in relation to the true total plant pixels selected from dataset.

### 4.6. Model Evaluation Metrics

To compare the performance of the MLP classifier with other methods, classical machine learning algorithms—random forest (RF) and support vector machine (SVM)—were employed to segment the plant images. RF is an ensemble learning method composed of a collection of multiple decision trees capable of modelling nonlinear variable interactions while avoiding overfitting. It evaluates an optimal set of decision trees that can analyze high-dimensional data. Each decision tree is developed using a set of data that does not depend on earlier decision trees. In this study, the probability distribution of each decision tree was aggregated and averaged to provide the final classification by labelling each pixel as 0 (background) or 1 (foreground). The SVM is a supervised learning method that is useful for the classification and regression of complex variables. In classification, an SVM model fits a hyperplane to a set of input data that separates one class from another. The best hyperplane is the one with maximum margin between a set of classes [60]. Generally, the SVM performs better and is more tolerant to irrelevant and interdependent features.

To assess the impact of feature selection on the performance of the models, six different models were developed using the AF and SF as inputs. The performance of the models was evaluated using the average accuracy score, F-score, and recall score metrics, as given in Equations (6)–(8):(6)$$\mathrm{Accuracy}=\frac{\mathrm{TP}+\mathrm{TN}}{\mathrm{TP}+\mathrm{TN}+\mathrm{FP}+\mathrm{FN}}$$
(7)$$R=\frac{\mathrm{TP}}{TP+FN}$$
(8)$$F\text{-}\mathrm{Score}=2\;\frac{P\;\times \;R}{P+R}$$
where P = $\frac{TP}{\mathrm{TP}+\mathrm{FP}}$; P is precision, R is recall, TP is true positive, TN is true negative, FN is false negative, and FP is false negative.

Each model used the entire dataset, which was divided into a 3:1:1 ratio for training, validation, and testing subsets, respectively. The validation subset was used to test the model during training, while the testing subset (independent data) was used to test the model after training. Appendix A shows the amount of data used for each feature dataset for training, validating, and testing the models. To ensure a robust and stable model, the training and testing procedures were repeated 100 times (with a random seed set to 0–99), and the results represent the average classification metrics for each set of data. Table 6 outlines the optimal hyperparameters used to train each model.

### 4.7. Assessment of Performance of SPAD Prediction Models

Three metric scores, namely the root mean square error (RMSE), adjusted determination coefficient (R^2^), and the mean absolute error (MAE), were used to comparatively assess the performance of the regression models. RMSE and MAE were used to evaluate the accuracy of the models, using the ground-truth SPAD data, while R^2^ evaluated the performance of the models based on the k-fold cross-validation [61]. The segmentation methods were labelled as MLP-FR, SVM-FR, RF-FR, ExG-FR, and ExGR-FR for field images and MLP-GR, SVM-GR, RF-GR, ExG-GR, and ExGR-GR for glasshouse data.

The normalized color vegetation indices (CVIs), which are commonly used in estimating plant chlorophyll and nitrogen, were extracted from the segmented images, and correlated (Pearson correlation) with the SPAD values. The dataset included 180 segmented cowpea images and 240 segmented wheat images, which were analyzed separately. Each dataset was divided into three portions in a 70:15:15 ratio for model training, and validation and testing. The training data were used to fit the model, while the validation dataset reduced overfitting during the fine-tuning of the hyperparameters [62]. The testing dataset provided an unbiased evaluation of the final model. The top five indices of each segmentation method were used as input features to train the respective regression models.

## 5. Conclusions

This study presented a novel method for automated and non-invasive segmentation of greenhouse and field plant images. Relevant information was extracted from sensitive features and used to train a one-layer MLP model that accurately distinguishes plant vegetation pixels from non-plant pixels. The model is capable of segmenting plants from different environments, ranging from glasshouses with simple backgrounds to field plants with cluttered backgrounds and varying illumination conditions. The selected features were also used to train classical ML models, SVM and RF, and their performances were compared to the proposed method. Although the classical ML models performed well, the MLP outperformed them in segmenting both glasshouse- and field-based images with high precision. The robustness of all segmentation methods was tested on plants from the glasshouse (cowpea) and the field (wheat) under different illumination conditions, and the proposed method produced the best performance in both cases. This method is not limited to segmenting green vegetation, but also plants with diseases and nutrient deficiencies, making it suitable for high throughput segmentation of plant images in both glasshouse and field environments.

## Figures and Tables

**Figure 1 plants-12-02035-f001:**
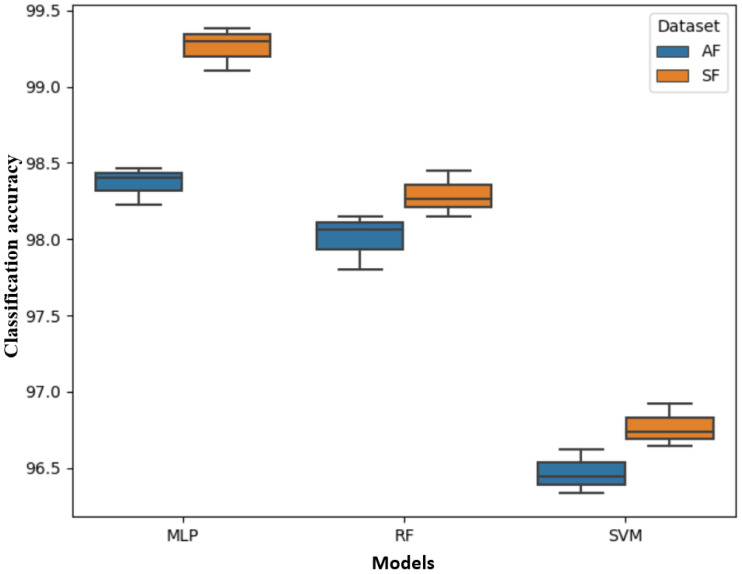
Classification accuracy scores for the three models (multilayer perceptron (MLP), random forest (RF), and support vector machines (SVMs) based on selected features dataset (SF) and all features (AF) dataset for training.

**Figure 2 plants-12-02035-f002:**
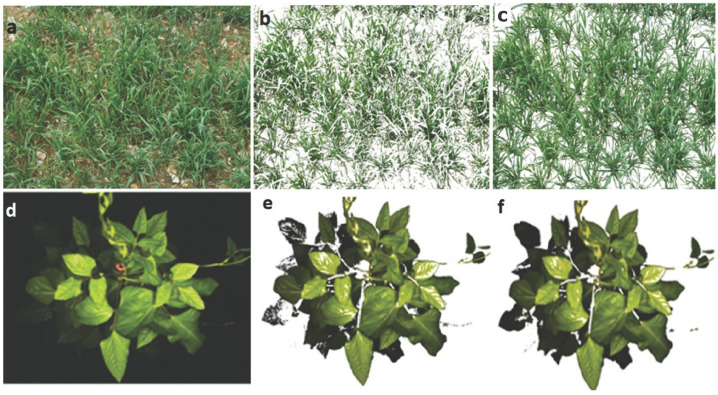
Examples of glasshouse and field segmented plants using the proposed method and selected segmentation methods. (**a**) Original wheat image, (**b**) ExG wheat segmented image, (**c**) proposed method (MLP) segmented image, (**d**) original cowpea image, (**e**) ExG cowpea segmented image, and (**f**) MLP segmented image.

**Figure 3 plants-12-02035-f003:**
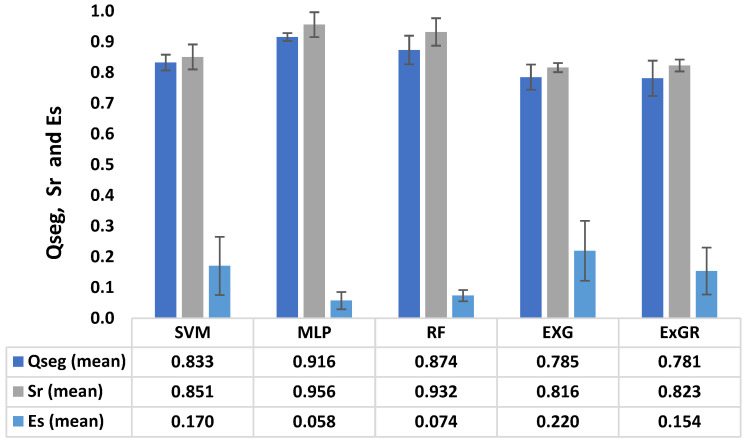
Mean segmentation accuracy rate comparison for quality assessment of five different segmentation methods for glasshouse-based images; Qseg measures the segmentation consistency on a pixel-by-pixel basis, Sr measures the consistency of plant pixels between image regions, and Es measures the rate of pixel misclassification. These were applied on the five segmentation methods: multilayer perceptron (MLP), support vector machines (SVMs), random forest (RF), excess green (ExG), and excess green–red (ExGR).

**Figure 4 plants-12-02035-f004:**
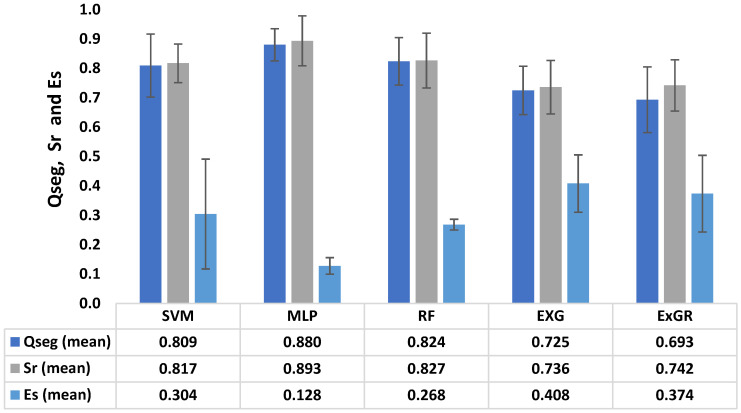
Comparison of segmentation accuracy rate (Qseg-, Sr, Es) for quality assessment of five different segmentation methods; multilayer perceptron (MLP), support vector machines (SVMs), random forest (RF), excess green (ExG), and excess green–red (ExGR) for field-based images. Qseg measures the segmentation consistency on a pixel-by-pixel basis, Sr measures the consistency of plant pixels between image regions, and Es measures the rate of pixel misclassification. These were applied to the five segmentation methods.

**Figure 5 plants-12-02035-f005:**
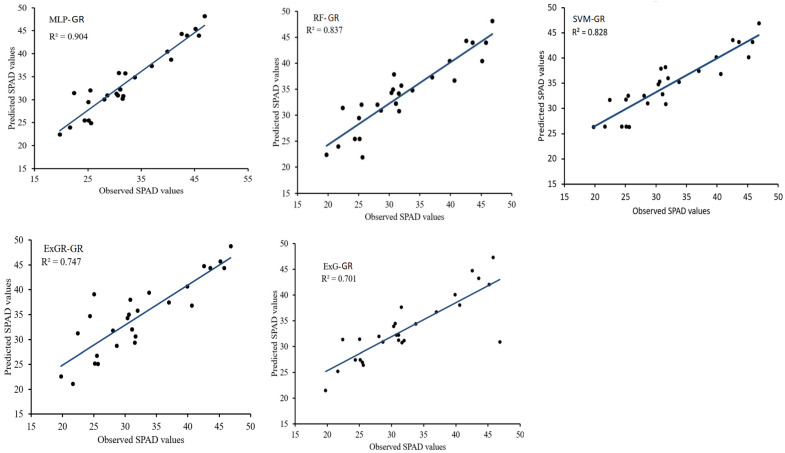
Scatterplots of predicted and observed SPAD values for the glasshouse plants. MLP-GR, RF-GR, SVM-GR, ExGR-GR, and ExG-GR represent the glasshouse-based regression models for the multilayer perceptron, random forest, support vector machine, excess green, and excess green–red segmentation methods, respectively.

**Figure 6 plants-12-02035-f006:**
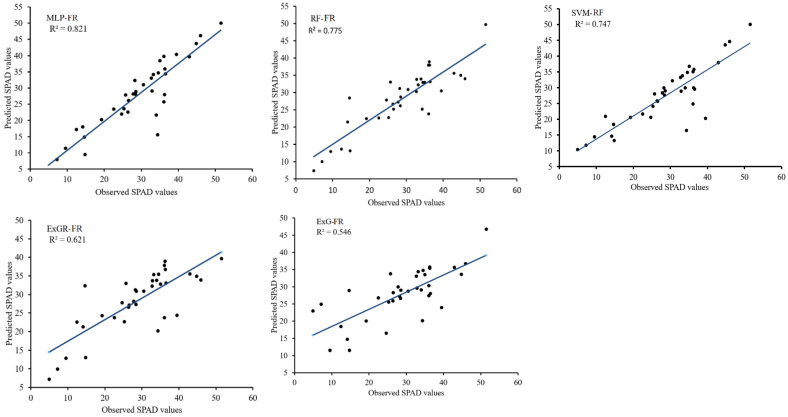
Scatterplots of predicted and observed SPAD values for the field plants. MLP-FR, RF-FR, SVM-FR, ExGR-FR, and ExG-FR are the field-based regression models for the multilayer perceptron, random forest, support vector machine, excess green, and excess green–red segmentation methods, respectively.

**Figure 7 plants-12-02035-f007:**
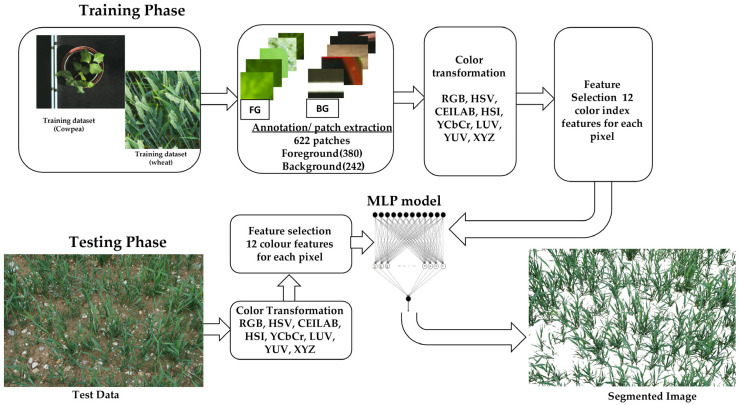
Diagrammatic representation of proposed method for glasshouse and field image, using MLP, multilayer perceptron.

**Figure 8 plants-12-02035-f008:**
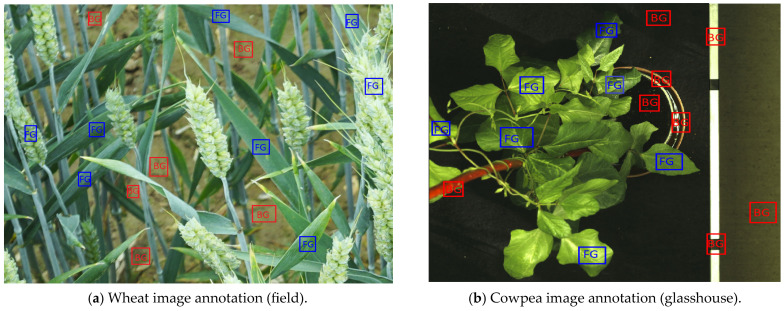
Annotation of images into foreground and background patches for feature extraction. (**a**) Wheat image annotation and (**b**) Cowpea image annotation. The FG represents the foreground annotation, and the BG is the background annotation.

**Figure 9 plants-12-02035-f009:**
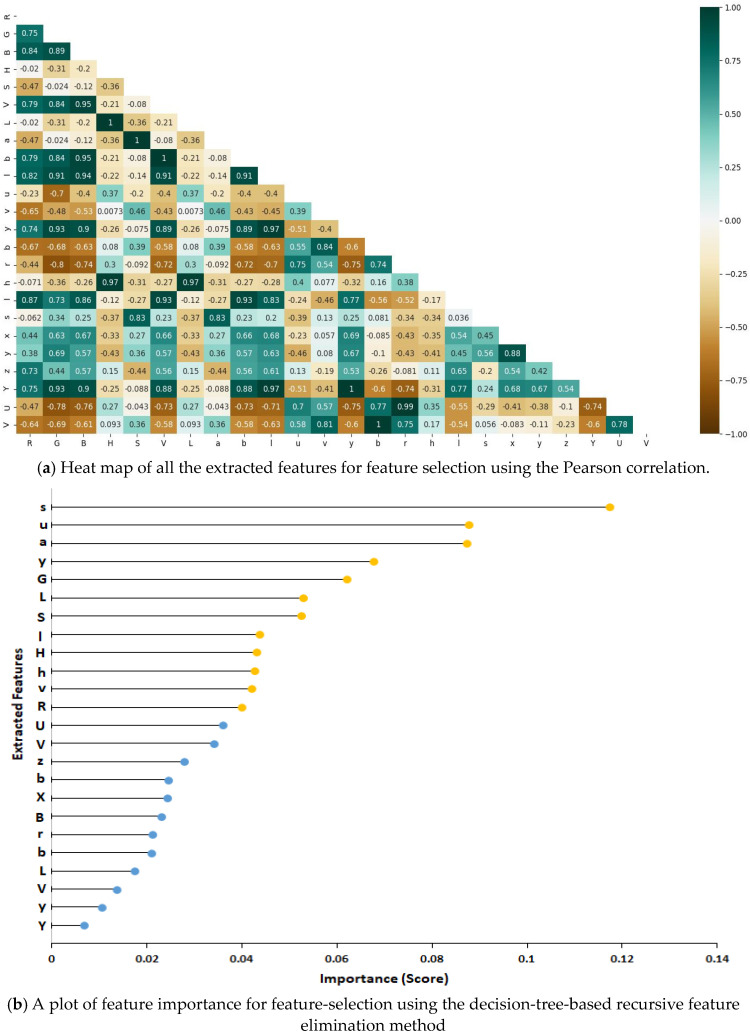
Feature-selection process involving correlation analysis and feature ranking based on importance score. (**a**) is the heatmap of all extracted features and (**b**) is a plot of feature importance for feature selection. Abbreviations: RGB (R = red, G = green and B = blue channels); HSV (H = hue, S = saturation, and V = value); ybr (y = luma, b = blue component, and r = red component); Lab (L = lightness, and a and b = chromaticity); YUV (Y = luma or brightness, U = blue projection, and V = red projection); Luv (L = luminance, u = blue axis, and v = red axis); hls (h = hue, l = lightness, and s = saturation); and XYZ (X and Z = spectral weighting curves, and Y = luminance).

**Figure 10 plants-12-02035-f010:**
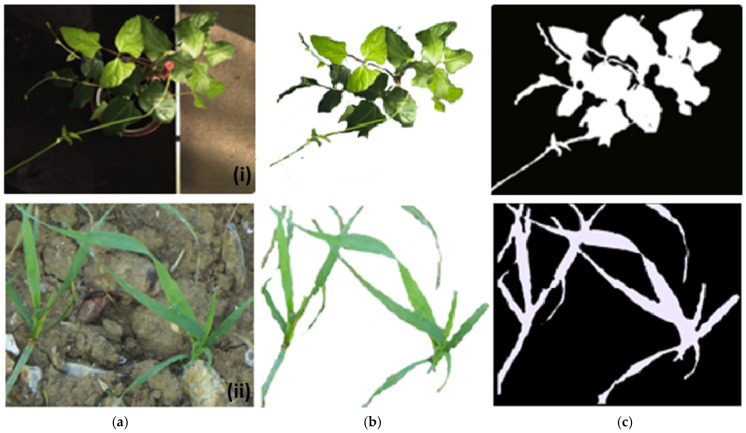
Examples of images obtained from the glasshouse and field, segmented as reference images. (**a**) Original image, (**b**) manually segmented image, and (**c**) binary image. Reference images were randomly selected from the dataset with different nutrient content and illumination and at variable growth stages.

**Table 1 plants-12-02035-t001:** Classification model performance assessment. MLP, RF, and SVM are the multilayer perceptron, random forest, and support vector machine segmentation models, respectively.

	All Features Dataset (AL)				Selected Features Dataset (SF)			
Learning Model	Accuracy	F-Score	Recall	Computational Time (s)	Accuracy	F-Score	Recall	Computational Time (s)
MLP	98.464 ^a^	96.142 ^a^	98.144 ^a^	175.098 ^a^	99.382 ^a^	97.423 ^a^	98.321 ^a^	113.521 ^a^
RF	98.011 ^a^	95.331 ^b^	97.421 ^a^	191.450 ^b^	98.354 ^ab^	96.632 ^a^	97.931 ^a^	128.455 ^b^
SVM	96.134 ^b^	92.402 ^c^	95.651 ^b^	560.560 ^c^	96.624 ^b^	92.822 ^b^	96.514 ^b^	321.502 ^c^

Superscripts ^a^, ^b^, and ^c^ represent significant differences with *p* < 0.05 within the columns. Same letters mean no significant difference, and different letters mean significant difference exist.

**Table 2 plants-12-02035-t002:** Correlation analysis of color vegetation indices (CVIs) and SPAD readings for cowpea. R, G, and B are the average red, green and blue values. The ExG, ExGR, MLP, SVM, and RF are the excess green, excess green–red, multilayer perceptron, support vector machines, and random forest segmentation methods, respectively.

CVI	Mathematical Expression	Correlation Coefficient (r)				
		Segmentation Method				
		ExG	ExGR	MLP	SVM	RF
Green	-	0.420	0.291	0.554	0.418	0.459
Blue	-	−0.228	−0.424	−0.682	−0.562	−0.605
ExG [21]	2 × G − R − B	0.742	0.763	0.893	0.824	0.838
ExR [7]	1.4 × R − G	0.692	0.822	0.885	0.745	0.820
CIVE [22]	0.441 × R−0.811 × G+0.385 × B +18.78	0.670	0.745	0.924	0.787	0.822
ERI [23]	(R−G) × (R−B)	−0.551	−0.568	−0.661	−0.517	−0.642
DGCI [24]	(H−60)/[60+(1−S)+(1−B)]/3	0.672	0.759	0.948	0.821	0.850
GR	G−R	−0.585	−0.721	−0.842	−0.766	−0.671
COM1 [17]	ExG+CIVE +ExGR	0.623	0.748	0.901	0.792	0.782
GBRG [21]	(G−R)/(R−G)	0.689	0.824	0.870	0.844	0.802
EGI [23]	(G−R)/(G−B)	0.231	0.425	0.464	0.382	0.295

**Table 3 plants-12-02035-t003:** Performance of the regression models during training and testing for glasshouse data. MLP-GR, RF-GR, SVM-FR, ExGR-GR, and ExG-GR are the glasshouse-based regression models for the multilayer perceptron, random forest, support vector machine, excess green, and excess green–red segmentation methods, respectively.

Model	Training Set			Testing Set		
	R^2^	RMSE	MAE	R^2^	RMSE	MAE
MLP-GR	0.942	2.152	2.120	0.904	2.682	2.931
RF-GR	0.881	3.885	2.581	0.837	3.958	3.780
SVM-GR	0.864	4.255	3.601	0.826	3.620	3.881
ExGR-GR	0.790	6.102	4.512	0.747	5.790	4.910
ExG-GR	0.772	6.881	5.155	0.701	7.151	6.851

**Table 4 plants-12-02035-t004:** Correlation analysis of color vegetation indices (CVIs) and measured SPAD readings for wheat. R, G, B, H, S, B are the average red, green, blue, hue, saturation, and brightness values respectively. MLP, RF, SVM, ExGR, and ExG are the segmentation methods using multilayer perceptron, random forest, support vector machine, excess green, and excess green–red segmentation methods, respectively.

CVI	Mathematical Expression	Correlation Coefficient(r)				
		Segmentation Method				
		ExG	ExGR	MLP	SVM	RF
Green	-	0.242	0.212	0.443	0.38	0.291
Blue	-	−0.377	−0.344	−0.618	−0.522	−0.552
ExG [21]	2 × G − R − B	0.704	0.726	0.927	0.841	0.877
ExR [7]	1.4 × R − G	0.675	0.722	0.845	0.751	0.803
CIVE [22]	0.441 × R −0.811× G+0.385 × B +18.78	0.601	−0.652	−0.842	−0.773	−0.782
ERI [23]	(R−G) × (R−B)	−0.407	−0.476	−0.681	−0.572	−0.644
DGCI [24]	(H−60)/[60+(1−S)+(1−B)]/3	0.623	0.690	0.881	0.791	0.747
GR	G−R	−0.551	−0.606	−0.740	−0.657	−0.671
COM1 [17]	ExG+CIVE +ExGR	0.532	0.682	0.813	0.722	0.78
GBRG [21]	(G−R)/(R−G)	0.690	0.743	0.904	0.781	0.811
EGI [23]	(G−R)/(G−B)	0.111	0.251	0.412	0.322	0.352

**Table 5 plants-12-02035-t005:** Performance evaluation of the regression models. MLP-FR, RF-FR, SVM-FR, ExGR-FR, and ExG-FR represent the field-based regression models for the multilayer perceptron, random forest, support vector machine, excess green, and excess green–red segmentation methods, respectively.

Model	Training Set			Testing Set		
	R^2^	RMSE	MAE	R^2^	RMSE	MAE
MLP-FR	0.893	3.25	2.52	0.821	3.680	3.233
RF-FR	0.815	4.05	2.78	0.775	3.785	3.781
SVM-FR	0.791	5.10	3.90	0.747	7.903	4.252
ExGR-FR	0.670	7.88	5.12	0.621	8.202	6.102
ExG-FR	0.642	9.85	7.55	0.521	13.155	11.851

**Table 6 plants-12-02035-t006:** Optimal hyperparameters for the different classification models.

Hyper Parameters	Model		
	MLP	SVM	RF
Hidden layers	1	-	-
Neurons	20	-	-
Activation function	ReLU	-	-
Kernel	-	RBF	-
C	-	0.001	-
γ	-	1	-
n_estimator	-	-	200
max_depth	-	-	20
min_samples_leaf	-	-	4

## Data Availability

The data used in this study are available from the corresponding author on request.

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
