# Peer review of "Machine Learning Methods for Automatic Segmentation of Images of Field- and Glasshouse-Based Plants for High-Throughput Phenotyping"

_plants, 2023, doi:10.3390/plants12102035_

Round 1

Reviewer 1 Report (New Reviewer)

Manuscript entitled  Machine Learning Methods for Automatic Segmentation of Images of Field and Glasshouse Based Plants for High Throughput Phenotyping” concerns the comparison of different methods responsible for segmentation of plant image and calibration of the model allowing for the evaluation of the SPAD parameter. In its current form, the manuscript is very difficult to read and understand the research concept. There are many fragments in the "Results" chapter that should be in "Materials and Methods". This makes it particularly difficult to understand how the model used for SPAD prediction was obtained and evaluated. There seems to be a lot of information missing about how many photos were used and what exactly was the procedure for taking them (how was the camera set up)

Author Response

Reviewer 2 Report (New Reviewer)

Research topic is very interesting and important. Manuscript is well written. The major issues I found is that the methods are not adequately described. So the work cannot be replicated.

1. Page 16: There is no explanation about how the decision tree-based recursive feature elimination works. It is required to either add detailed description of the algorithm, or add a reference and some brief description of the method, if this has been previously published.

2. Page 18, Artificial neutral network: Mathematical formula (7) was not well explained. What does the activation function ah and a0 exactly look like? There is little details about how the computation is done. It mentions back and forward propagation algorithms. But again there is no detailed description or references. 

3. Page 19: XGBoost: Again, it lacks description of this approach. Why is this approach was chosen? What is the advantage of this approach compared to other regression methods? 

Author Response

Reviewer 3 Report (New Reviewer)

The present manuscrip test a new machine learning method to discriminate plants from background in field and greenhouse conditions and also able to predict the SPAD status of the plants.

The document is well written, the results are clearly presented with a good discussion. The only major thing that I want to highlight is that in the results section there are two section where the main content is still a description of the method used. These parts should be moved to the M&M section and reduced them to a short sentence in the results is more convenient for the lecture of the section.

I also would like that in all the figures and tables for all the the acronyms present it should be included the meaning in the legend or in footnotes.

If not there are some typing errors that I have highlighted in the attached documents.

Round 2

Reviewer 1 Report (New Reviewer)

The work has been significantly improved and is ready for publication

Reviewer 2 Report (New Reviewer)

All my previous comments have been sufficiently addressed. 

This manuscript is a resubmission of an earlier submission. The following is a list of the peer review reports and author responses from that submission.

Round 1

Reviewer 1 Report

1. Ref 2 is incomplete. re-check all the references

2. How robustness of the proposed algorithm can be proved? Give explanation. 

Reviewer 2 Report

I suggest to present the methodology earlier than the experiment section.

The method design is simple, which is good; however, how about considering bigger models like ResNet, DenseNet. 

The related work is also not comprehensive. Many image segmentation methods are missed, for example, Rethinking Semantic Segmentation: A Prototype View, Regional Semantic Contrast and Aggregation for Weakly Supervised Semantic Segmentation. 

Reviewer 3 Report

Title: Machine Learning Methods for Automatic Segmentation of Images of Field and Glasshouse Based Plants for High Throughput Phenotyping

Overview: This manuscript presents a method to phenotype wheat and cowpea plants in greenhouse and field using images to predict SPAD (Soil Plant Analysis Development). The proposed method implements MLP, SVM, RF, ExG, and ExGR as ML models and the results look great.   This manuscript like many other manuscripts use machine learning models, methods, and techniques as a tool to segment RGB images of plants for phenotyping. The  manuscript falls very heavy on the application side of machine learning and uses standard, and commonly used methods to achieve what is already abundantly reported in the literature. phenotyping is an important problem in agriculture, and commendable effort, I'd encourage the authors to go beyond just pure application to tackle these real world problems.   Furthermore, deep learning based methods with end-to-end learning are outperforming all classical methods reported in this works. Their methods, metrics such as IOU and other techniques and their limitations could provide more motivations.    The format of the manuscript, unless this is a journal specific format or a commonly accepted norm in this field, the organization of having results in later and materials and methods upfront on the manuscript seems a bit natural and provides better flow for reading and understanding.

Specific Comments:

Abstract:
Like SVM and ExG, SPAD and MLP could also be abbreviated to make it clearer
Introduction:
# 56-58.. Reference required
# 67-69 "The supervised mean-shift algorithm, k-means
clustering and fuzzy clustering are commonly used learning-based methods for plant seg-
mentation [7]." is s wrong statement. Either coming from the authors or the reference. these are unsupervised, especially kmeans
  A table that summarizes all data and no. of images for training and testing would be helpful. Currently, it is confusing. e.g. line 124-126. "For each model, the entire dataset was first divided into train and test subsets in an 80%-20% ratio, respectively. The test subset (not used during the training) was used to validate the model after training."   the last statement on test subset was not clear. after 80-20 division, where does this extra data come from?   Results:

# 125-126 "The test subset (not used during the training) was used to
validate the model after training" not clear... what was 80-20 distribution

Table 2: highlight most significant in each column so its clear to see which one has the best outcome   Also, if the computation time is in hundreds of seconds, how is this high throughput?   Some confusion about glasshouse and field images on wheat vs cow peas. Was all wheat plant images from the field and all cow peas from the glass house? Also, is glass house and green house the same?